# Effect of Demographics and Time to Sample Processing on the qPCR Detection of Pathogenic *Leptospira* spp. from Human Samples in the National Reference Laboratory for Leptospirosis, Brazil

**DOI:** 10.3390/tropicalmed8030151

**Published:** 2023-02-28

**Authors:** Romulo Leão Silva Neris, Mariana Cristina da Silva, Mariana da Silva Batista, Keila de Cássia Ferreira de Almeida Silva, Ilana Teruszkin Balassiano, Kátia Eliane Santos Avelar

**Affiliations:** Laboratório de Referência Nacional para Leptospirose/Centro Colaborador da Organização Mundial da Saúde para Leptospirose, Instituto Oswaldo Cruz, Rio de Janeiro 3304557, Brazil

**Keywords:** leptospirosis, human samples, qPCR, diagnostics, DNA extraction

## Abstract

Leptospirosis diagnosis by MAT requires antibody levels that are typically present only after the first week of symptoms, many days after infection. To improve testing capacity and to develop a fast and reliable solution for the diagnosis of this disease in the first few days after clinical manifestations, the National Reference Laboratory for Leptospirosis/WHO Collaborating Center in Brazil implemented a duplex molecular method by qPCR for human samples for the detection of the gene *lip*L32, conserved in pathogenic *Leptospira* spp. In this paper, we describe the overall performance of this protocol in the first 3 months as a standard routine. Detection of pathogenic *Leptospira* spp. DNA was similar between blood, plasma, and tissue samples, with a limit of detection as low as one cell per sample, and among 391 samples from suspected cases, 174 (44.6%) were positive. The average *RNASEP1* control gene detection cycle thresholds (Ct) were 28.4 and 29.8 for positive and negative samples, respectively. The median sample collection interval from the beginning of symptoms was 3 days for positive and 4 days for negative samples, respectively. Neither age, sex, nor the time intervals between sample collection and DNA extraction significantly influenced the results. Surprisingly, positivity was related to the time between DNA extraction and the qPCR reaction. These data support the use of this routine as a diagnostic approach to strengthen the molecular detection of leptospirosis and to develop new strategies.

## 1. Introduction

Leptospirosis typically manifests itself after an incubation period of two days to more than two weeks [1]. Infected patients seek healthcare after the onset of these symptoms, and it is during this time window that a diagnosis is determined. The gold standard for this disease’s diagnostics is the microscopic agglutination test (MAT), which suggests infection based on the detection of antibodies responsive to a set of reference antigens [2]. However, an important limitation of this technique is that the increase in antibody titers in serum usually happens after the end of the first week of symptoms, with a posterior stabilization during convalescence [3]. Due to this limitation, the serological diagnosis of samples collected immediately after the beginning of symptoms favors the occurrence of false-negative results.

Another problem is that multiple guidelines recommend, ideally, the evaluation of MAT results based on at least two temporal-spaced samples to confirm an acute infection, with a relative increment in the antibody titer between samples or with its relative reduction, if one of the samples was collected after convalescence [4,5,6]. However, in emergency situations, such as floods, or in regions with poor access to healthcare, it is really challenging to obtain two or more timely-spaced samples from a suspected case.

The interval between symptom onset and sample collection includes the time it takes for the person to reach a healthcare facility, time it takes for evaluation, and time it takes to mobilize a healthcare worker, when available, to collect blood and plasma. The interval from sample collection up to DNA extraction comprises the time it takes to pre-process or split the sample into smaller aliquots if it is going to be sent to multiple labs for differential diagnosis, the time it takes to provide appropriate conditioning for the sample before transport, the time needed to transport the sample to the reference lab for testing, and the period for quality control on the sample prior to DNA extraction. The last major steps are in the lab, where the sample is going to be tested, the DNA will be extracted, and it will undergo proper conditioning prior to the qPCR reaction and its analysis. During each one of these steps, parameters such as freeze/thaw cycles and sample handling can lead to a reduction in sample quality and potentially increase the occurrence of false-positive and false-negative results.

To ensure a reliable method to detect infections at the beginning of the disease with one sample available, the National Reference Laboratory for Leptospirosis/WHO Collaborating Center implemented a standard routine of qPCR to replace the previous conventional PCR detection of the *lip*L32 gene of pathogenic *Leptospira* spp. that was described by other authors previously [7,8]. This routine was implemented during the floods that occurred in Brazil during the first semester of 2022 in the cities of Petrópolis/RJ and Recife/PE, and it greatly increased the overall capacity of the laboratory. In this work, we report our findings on the effects of the different parameters on *lip*L32 gene detection.

## 2. Materials and Methods

### 2.1. Study Setting

At the beginning of 2022, the National Reference Laboratory for Leptospirosis (NRLL)/World Health Organization Collaborating Center (WHO-CC) in Brazil implemented the diagnostic routine for leptospirosis by qPCR, replacing the existing one, based on a conventional PCR approach. Samples used for diagnosis were compared to other parameters to measure the reliability of the method and identify critical factors that could impact the diagnosis of this disease.

### 2.2. Samples Acquisition and Conditioning

Blood, plasma, sera, and tissue samples belonging to the sample bank of the NRLL/WHO-CC were used. This sample bank was registered under the ethics committee number 5,277,660, following Brazil’s CONEP regulations. Three hundred and ninety-one (391) samples were obtained between 1 January and 15 May 2022, from the cities of Petrópolis/RJ and Recife/PE; they were sent to the laboratory by health facilities and Brazilian central labs for serological diagnosis and stored after the release of the results. Samples with descriptive information about sex, age, and time from collection after the onset of symptoms were included in this study. Non-labeled data were obtained and used for retrospective analysis.

Up until DNA extraction, all samples were kept at −20 °C indefinitely. After extraction, the DNA obtained was also stored at −20 °C until it was used in the qPCR reaction.

### 2.3. DNA Extraction

From each sample, pure DNA was obtained using silica-based affinity column purification according to the manufacturer’s specifications. For sera, plasma, blood, and culture samples, DNA was obtained starting with a volume of 200 µL of sample using the QIAmp DNA Mini Kit (Qiagen, Hilden, Germany). For tissue samples, 25 mg of tissue were used to obtain DNA from the DNeasy Blood & Tissue Kit (Qiagen, Hilden, Germany).

### 2.4. Duplex Quantitative Real-Time PCR

Reactions to detect *Leptospira* spp. DNA were carried out by a duplex reaction. In this reaction, the *lip*L32 gene amplification was targeted by a probe-based set of primers, and in the same reaction, the *RNASEP1* gene was used as an endogenous positive loading control. For this, the reaction was performed using the QIAGEN Quantinova Probe RT-PCR MasterMix (Qiagen, Hilden, Germany) according to the manufacturer’s instructions. Primer and probe sequences were prepared as first described by Riediger et al. [7], with differences including a designed modification from FAM fluorophore to a Cy5 at the 5′ of the *RNASEP1* probe and the replacement of Black Hole Quencher 1 (BHQ-1) for BHQ-2 at the 3′ end as the quencher of the probe on the same primer, as shown in the following Table 1:

RNAse-free water (Qiagen, Hilden, Germany) was used in the reactions, and all sets of reactions carried a *lip*L32 positive control from DNA extracted from a *Leptospira interrogans* culture with densities of 10^6^ up to 10^7^ cells/mL, an *RNASEP1* positive control from human sera confirmed to be negative for leptospirosis, and a blank reaction from RNase-free water (Qiagen, Hilden, Germany) instead of DNA. The qPCR reaction was performed on a Rotor-Gene Q thermocycler (Qiagen, Hilden, Germany), and the Ct was determined automatically by the equipment software QIAGEN Rotor-Gene Q (Qiagen, Hilden, Germany, version 2.3.5) using negative and positive controls as internal references for each assay. In the analysis of sensitivity curves and the determination of the detection threshold, samples were run as duplicates. For the routine diagnosis, each sample was analyzed without replicates.

The following qPCR criteria were used to analyze the data, as shown in Table 2: The experiments were first analyzed to determine if *lip*L32 and *RNASEP1* control detections were considered valid; if these criteria were met, each sample was considered valid if the *RNASEP1* gene was detectable in that sample. If yes, detection of *lip*32 was used to determine positive and negative samples.

### 2.5. Leptospira *spp.* Culture

The *L. interrogans* serovar Copenhageni M20 reference strain (CLEP 00002) was obtained from the *Leptospira* spp. Collection (CLEP-Oswaldo Cruz Foundation, Rio de Janeiro, Brazil) and grown at 28 °C (7–10 days) in Ellinghausen–McCullough–Johnson–Harris (EMJH; Difco, Isère, France) broth supplemented with 1% bovine serum albumin (Sigma, St. Louis, MO, USA). Cultures were maintained routinely. To determine the cell density of the culture, seven-day grown cultures were counted under a Petroff-Hausser chamber, and 2 × 107 bacteria/mL were used for the following steps.

To obtain *Leptospira* spp. DNA curves from cell cultures, pure DNA was extracted from ten-fold serial dilutions of 106 cells up to 0.1 cells. *Leptospira* spp. DNA was analyzed for the *lip*L32 gene on the same day of extraction, without freezing or other methods of storage. After that, the samples were kept at −20 °C indefinitely.

### 2.6. Data Analysis and Statistics

The date of the onset of symptoms and other information were collected from the patient’s history after the patient’s admission to the hospital by filling out the epidemiological form. Before the analysis, all sensitive information about the patients, except for race, age, and sex, was discarded, and non-labeled data was shuffled in Excel to avoid identification.

Statistical analysis graph design was performed by GraphPad Prism 8.0 (GraphPad, Dotmatics). The criteria for each analysis are described in the appropriate result. In all analyses, a *p*-value of at least 0.05 was considered to be significant to the phenomenon being evaluated.

## 3. Results

### 3.1. Sample Demographics and Time Intervals from Sample Collection up to qPCR Reaction

Between January and May 2022, 391 samples from suspected cases were tested. These samples included 363 serum samples, 22 blood samples, 3 tissue fragment samples, 2 DNA samples, and 1 plasma sample. As reported in Table 3, among the 391 samples included in this study, 250 (63.94%) were from male patients, 140 (35.81%) were from female patients, and 1 (0.26%) was from a patient whose sex was not declared. From these samples, the median age was 36 years, with first and third quartile ranges (1st–3rd IQR) of 24 and 49 years, respectively.

In this population, most patients self-declared at admission as being white (128; 32.74%); 68 (17.39%) self-declared as being brown; 40 (10.23%) declared themselves as yellow; and 33 declared themselves as black (8.44%). The races of a large number of patients were not reported (122; 31.20%). This could be due to incomplete registration during admission or the patients’ decision not to inform the hospital of any race at the time of admission.

Using the date of the symptoms’ onset as a starting point, we traced time intervals up to the moment when the qPCR reaction was realized to provide a diagnosis, and these intervals are reported in Table 4. Among the samples with adequate registration (91.04% of all samples), the median time from the onset of symptoms up to the sample collection date was 4 days (1–6 days for 1st–3rd IQR); the time interval of 389 samples (99.49%) from sample collection up to the DNA extraction was 9 days (7–14 days for 1st–3rd IQR), the largest time interval among intervals relevant during diagnosis; and the median time from DNA extraction up to the qPCR reaction from 387 (98.98%) samples was 3 days (1–5 days for 1st–3rd IQR).

Together with the results obtained for *lip*L32 and *RNASEP1* detection, the effects of each of the demographic and time interval variables were determined to understand how the diagnosis and leptospirosis prevalence can be related.

### 3.2. lipL32 and RNASEP1 qPCR Detection among Tested Samples

Among these 391 tested samples, the *lip*L32 gene was detectable in 174 samples (44.50%) and not detectable in 207 samples (52.94%) (Figure 1A). Among these samples, 10 were inconclusive for the detection of *lip*L32, all of them due to the failure to detect the *RNASEP1* control gene. The levels of detection of *RNASEP1* were compared between detected and non-detected *lip*L32 samples to assess the reliability of the detection from the DNA extraction efficiency. While the average *RNASEP1* gene Ct for detectable samples was 28.44, the average *RNASEP1* gene Ct for non-detectable samples was 29.75 (*p* < 0.001, Figure 1B). This data suggests that *lip*L32 detection is dependent on *RNASEP1* gene detection efficiency. To further investigate if this finding could impact the Ct variability of *lip*L32 in detected samples, we performed a linear regression comparing the Cts of both genes. The R2 in these samples was 0.0036, with no significant difference in *lip*L32 Ct with an *RNASEP1* variation (Figure 1C). This data points out that *Leptospira lip*L32 gene Ct determination was consistent among detectable samples, even with a higher *RNASEP1* Ct.

Next, we investigated the time needed for the collection of the samples using the results. Among 189 qPCR positive results and 157 negative results with reported times of sample collection (356 total), we found (Table 5) a median of 3 (1–6 days for 1st–3rd IQR) for positive samples and 4 for negative samples (1–7 days for 1st–3rd IQR). However, time differences among these two groups were not significant (unpaired t test, CI 95%, *p* = 0.1097).

### 3.3. Effects of Demographics in lipL32 Detection from Human Samples

After that, we evaluated the impact of age, self-declared sex, and race in the data obtained from *lip*L32 gene detection. Stratification of sex for this gene detection by qPCR showed no significant differences in the average proportion of samples with a detectable Ct, with a positivity rate of 46.47% for men and 44.29% for women in the group (*p* = 0.6804, Figure 2A).

To understand if age was an important factor in the diagnosis of leptospirosis by qPCR, the average age of patients with a positive detection of the *lip*L32 gene (38.2 years) and the average age of suspected cases without the detection of the gene (35.9 years) were compared. There were no significant differences between these groups (*p* = 0.2205, Figure 2B). The data was also compared to the age distribution of patients in the tested samples (Figure 2C), and no differences in positivity rates were found (*p* = 0.3592).

Next, we accessed the race distribution of patients’ samples and tested them to check if demographics could be correlated with the detection rate among those samples. Interestingly, we found different frequencies of detection among self-declared races, with the highest positivity by qPCR being with self-declared brown patients (66.16%), followed by white patients (28.91%), black patients (36.36%), and yellow patients (20%). Among all samples with no declared race, the positivity was 59.01% (Table 6). The association between race and positivity was statistically significant, with Χ^2^(3) = 32.97, *p* < 0.001.

### 3.4. Effects of Samples Processing Time in lipL32 Detection from Human Samples

For this part, we analyzed the positivity correlation among samples and the intervals from the onset of symptoms and collection (T1), in between collection to DNA extraction (T2), and from DNA extraction up to the qPCR reaction (T3) (Figure 3A).

Correlation analysis showed that both T1 and T2 time intervals had a poor correlation with *lip*L32 detection (−0.09 and 0.03, respectively). These were the largest intervals in our DNA processing procedure, with medians of 4 and 9 days, respectively (Table 4). Despite having the smallest median interval (3 days), T3 was responsible for the strongest effect in this correlation (−0.18), suggesting that increases in the interval to perform a qPCR assay after extraction could lead to an increase in the expected proportion of *lip*L32-negative samples (Figure 3B).

## 4. Discussion

In this work, we presented a comprehensive analysis of demographics and other factors and their correlation with leptospirosis’ diagnosis by *lip*L32 gene detection by qPCR using a duplex assay. This analysis was generated from the data of samples for which molecular diagnosis was requested to be performed in the National Reference Laboratory for Leptospirosis/WHO Collaborating Center in Brazil. First, the detection threshold was accessed using serial dilutions of the reference strain *L. interrogans* serovar Copenhageni M20 culture alone and mixed with plasma and blood. In these preparations, we found a threshold of 34–35 Ct. Riediger, 2017 [7] found a Ct of 38.44 when proposing these gene detections to diagnose leptospirosis by qPCR for a minimum amount of 1 genomic equivalent per reaction. This difference is acceptable mainly because our analysis was based on the detection of *lip*L32 per serial dilution of cells, not its genomic equivalent. The lack of plasmids or other control DNAs for this gene is a limitation for properly comparing both approaches. However, using cell number instead of the genomic equivalent is related to the diagnostic routine because both MAT cultures are used in a defined cell concentration, and the isolation detection method is based on the presence of viable *Leptospira*, with a countable cell number under microscopy [9]. This can also facilitate the implementation of qPCR diagnostic routines for regional and other labs without easy access to plasmids or other purified DNA controls.

qPCR leptospirosis diagnosis is more reliable when performed in samples prior to the first seven days of symptoms [4,7,10], and MAT diagnosis is usually recommended after the second week of symptoms [9,11]. One limitation for this analysis was that, usually, authors compare the efficiency of both MAT and qPCR, but this is mainly performed to demonstrate the complementarity of both techniques in a lab for an appropriate diagnosis of leptospirosis [12,13,14,15]. Philip et al. [12] showed that less than 25% of qPCR-positive samples are also MAT-positive samples. Other reports show a similar level of correlation for both techniques. With this, we decided not to use MAT data to strengthen the findings, and since this analysis was retrospective, we only had access to the standard qPCR test for samples where the criteria were met and qPCR was requested. In this work, for qPCR *lip*L32 detection, we found a lower median of symptoms’ time of onset in positive versus negative results. It is expected that the qPCR detection is more sensitive in the first days of disease; however, we did not find significant differences in this distribution. This could be due to the very similar collection time for samples that are primarily designated for the qPCR diagnosis of leptospirosis, reducing an effect that is more commonly reported with a difference of weeks in sample acquisition [16,17,18]. It is important to consider the impact of confounders that can increase the trend toward false-positive or false-negative results but are not directly related to the pathogenesis mechanisms, such the region of the sample and the socioeconomic status of the analyzed sample cluster [19,20]. The probable main reason is that the concurrence of certain conditions increases the risk of infection in a certain group, increasing the likelihood of detection when we concentrate samples with these characteristics, such as during environmental disasters, where we usually have to test hundreds of samples from the same place in a short period, in comparison with seasonal routine activities to test suspected cases, where we test samples all over the country but in a lower number. To overcome this, researchers can design statistical analyses with larger sample sizes to generate representative descriptions of populations. Another way to increase the accuracy of the test in reference laboratories is to use well-designed, standard operational procedures to control sample collection, handling, and storage.

In the period analyzed in the paper, we found a positivity rate of 44.5% among tested samples for suspected cases. The high prevalence among these samples can be related to two main reasons: first, the samples were collected in a period of seasonal increase in tropical floods and storms, which increases the number of related febrile illnesses over its duration [21]. Second, testing only suspected cases, we had a trend toward increasing the rate, in comparison with larger observational studies based on serological surveys [22,23,24]. Important findings here are that the detection of the *lip*L32 gene using this system was only slightly affected by overall DNA quality based on *RNASEP1* gene detection (R^2^ = 0.0036). This points out that this system is robust and appropriate to be used as a standard routine, even in scenarios that would reduce DNA quality or cause poor efficiency in DNA extraction. Despite leptospirosis having a different incidence among male and female humans according to literature, *lip*L32 gene positivity was not influenced by sex in this paper [25,26]. Regarding age effects on this gene detection by qPCR, except for 0–9-year-old suspected patients, we saw no effects of age on the positivity. The zero-to-nine-year-old effect of lower positivity could not be fully considered, especially due to the lower sample size in comparison with other age groups (N = 16 for the 0–9 years old average age group sample size = 42.77 years).

In this work, we also found a significant effect of race on the positivity rates among suspected cases, with a trend for brown people’s samples to test positive for *lip*L32. It is important to note that, despite many guidelines addressing how to approach race, color, and ethnicity in biological studies in different ways, we chose to include this information exactly as it is called and registered by the Brazilian Public Healthcare System [27]. We must consider that this information is self-declared and can be biased towards the perception of each patient according to their region. Other studies have shown differences in the overall incidence of leptospirosis by self-declared race and other factors, such as income and access to healthcare [28,29,30]. In Brazil, we have a high correlation between race and social inequality, which hinders access to basic public services, such as treated water supplies, waste management, disaster management, and access to adequate healthcare, including diagnosis. Therefore, it is possible that the difference observed in race groups is directly related to the different distribution of access to these services, although we do not have the tools to stratify and analyze this finding specifically, keeping this question to be answered in other socio-economic studies.

During routine diagnostics, timing is a critical step to ensure a proper diagnosis and the confirmation of a suspected case. A rapid response is required, especially in leptospirosis, where disease progression can be as fast as 48 h [31]. The detection of pathogenic *Leptospira* is important to describe the increase in cases in a specific region after a flood. DNA is a very stable biomolecule, even in non-physiological conditions, such as high and low temperatures [32]. In reference diagnostics, tested samples are also affected by seasonality and demand. A larger number of samples are expected to arrive during floods and storms, due to the increased chances of humans coming into contact with potentially infected water [28]. Therefore, it is important to understand if *lip*L32 detection is stable among samples with different collection time intervals. We expected that longer time intervals from the onset of symptoms up to the qPCR reaction could increase the rate of false-negative assays if the number, duration, or conditioning during each step led to a reduction in DNA quality. In this work, we showed that, among all time intervals analyzed, the interval from DNA extraction to qPCR reaction is the most critical for the detection of the *lip*L32 gene. This was interesting because this interval is the only part of the diagnostic process that we can track and evaluate. We expected T1 (symptoms/collection) and T2 (collection/extraction) to have a stronger effect than T3 (extraction/qPCR) because we received samples from multiple states and hospitals for extraction and analysis, which were manipulated by different professionals using different protocols for collection and different conditioning and transportation until the extraction. Despite that, having this interval as the one with the largest effect that correlates with the result makes it easier to overcome in the routine than parameters from the other intervals. Our group has been decreasing the time from DNA extraction to qPCR reactions, aiming to perform this last step in less than 24 h (median) to evaluate the impact of positive tests in future reports.

## 5. Conclusions

This work showed that *lip*L32 gene detection is a reliable protocol that can be used as a reference diagnostic tool for leptospirosis. The duplex reaction ensures an increase in analysis’ speed, lowers costs per sample, and is a powerful tool, especially during high-demand times, such as emergency periods. Most of the entry parameters were eliminated as confounding or correlating factors with the detection. More studies are needed to understand the impact of the time interval between DNA extraction and reaction and the differences in positivity among self-declared races.

## Figures and Tables

**Figure 1 tropicalmed-08-00151-f001:**
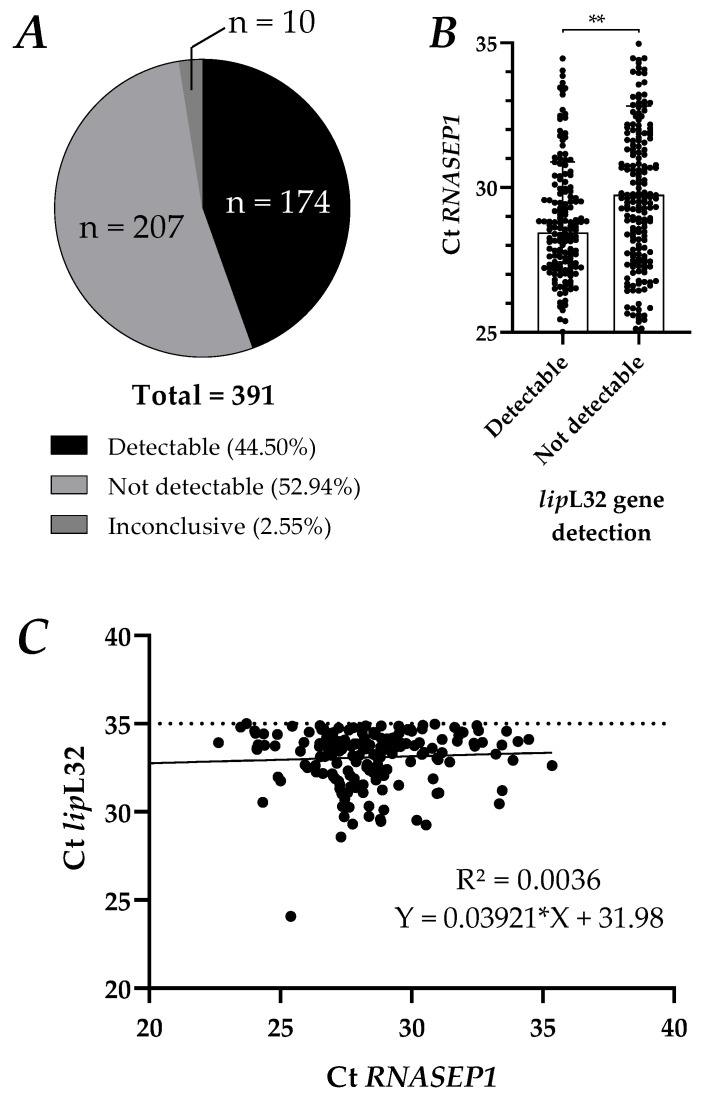
Data from human samples tested for *lip*L32 gene detection in a diagnostic routine by qPCR. DNA from 391 human samples of leptospirosis-suspected cases was extracted and then analyzed for the detection of *lipL*32 and *RNASEP1* genes using duplex qPCR. (**A**) A representation of the number (n) of samples found to be detectable, not detectable, or inconclusive in the panel. (**B**) the *RNASEP1* Ct for both detectable and non-detectable samples; the number in parentheses represents the fraction of each diagnostic result among total samples. Each dot is an independent sample. Bars show the average + S.D. An unpaired, two-tailed t test was used to perform the analysis; ** *p <* 0.01; (**C**) linear regression of *lip*L32 Ct only from detectable samples versus the respective *RNASEP1* Ct; R2 was reported on the graph, along with the predicted f(X) equation. The dotted line represents the detection threshold for detectable samples, and the continuous line is the regression trendline.

**Figure 2 tropicalmed-08-00151-f002:**
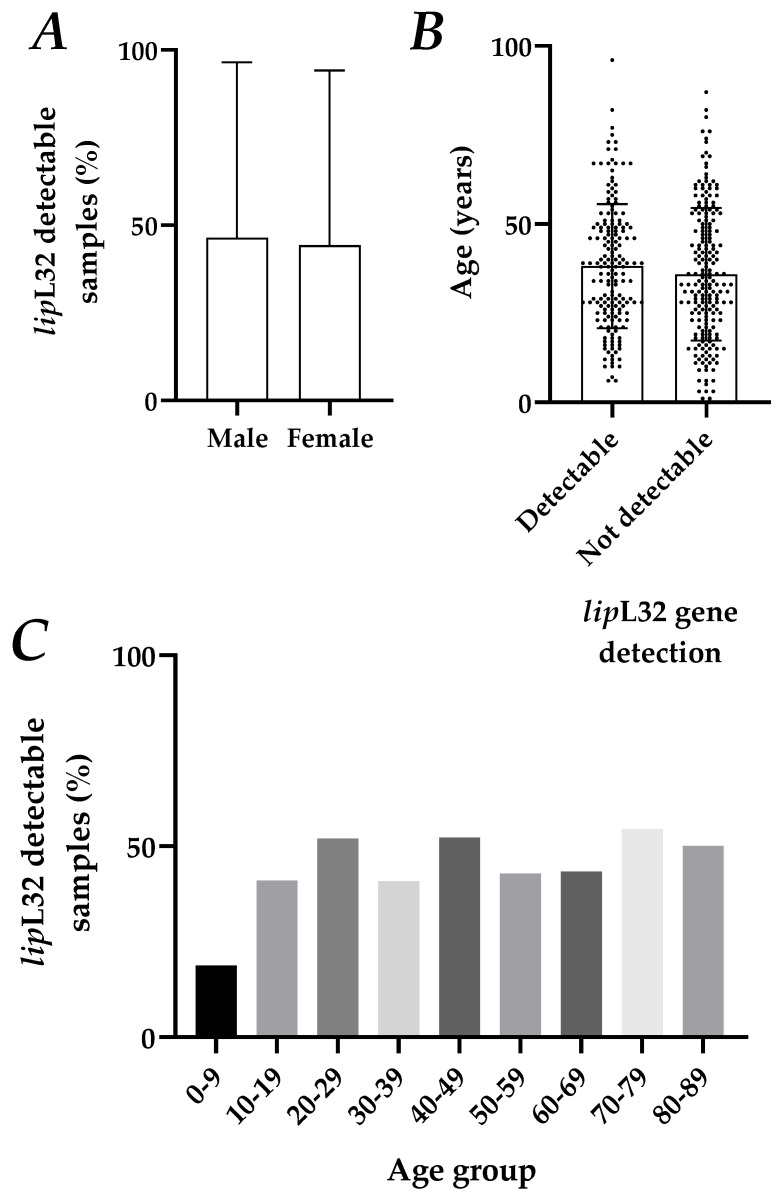
Data from human samples tested for *lip*L32 gene detection in a diagnostic routine by qPCR. DNA from 391 human samples from leptospirosis-suspected cases were extracted and then analyzed for the detection of the *lip*L32 and *RNASEP1* genes by duplex qPCR. (**A**) A representation of the proportion of samples found to be *lip*L32-detectable among male and female patients is shown in the panel. An unpaired, two-tailed t test was used for the analysis; (**B**) age of patients with detectable and undetectable *lip*L32 gene amplification by qPCR; each dot represents an independent sample. Bars show the average + S.D. The analysis was performed by an unpaired, two-tailed t test; (**C**) age distribution of *lip*L32 gene detection (%) among samples. A one-way ANOVA test was performed.

**Figure 3 tropicalmed-08-00151-f003:**
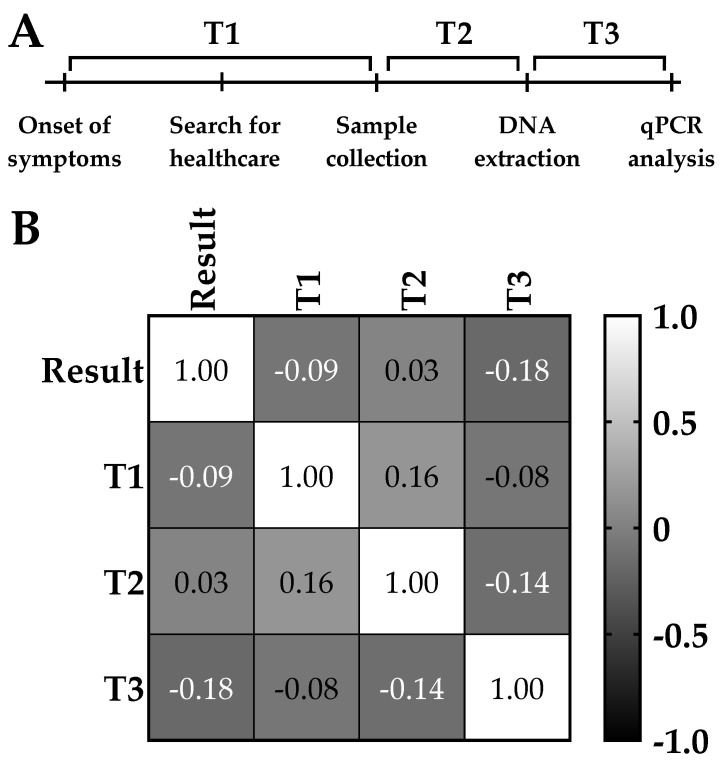
Correlation plot of *lip*L32 gene detection by qPCR and the time intervals from sample collection up to the reaction. For each sample, “Result” is either 0 (not detected) or 1 (detected). T1 denotes the time (in days) between the onset of symptoms and collection, T2 the time between collection and DNA extraction, and T3 the time between DNA extraction and the qPCR reaction. DNA from 391 human samples from leptospirosis-suspected cases was extracted and then analyzed for the detection of *lip*L32 and *RNASEP1* genes by duplex qPCR. In the panel, (**A**) the timeline of main events was used to determine time intervals and (**B**) correlation was performed using the Pearson coefficient for every pair of datasets. For these data, the P-value was determined by a two-tailed, CI 95% analysis.

**Table 1 tropicalmed-08-00151-t001:** Primers and probes used in this study.

Primer/Probe	Sequence	Final Concentration
*lip*L32 Forward	5′-AAG CAT TAC CGC TTG TGG TG-3′	200 nM
*lip*L32 Reverse	5′-GAA CTC TTT CAG CGA TT-3′	200 nM
*lip*L32 Probe	5′-/56-FAM/AAA GCC AGG ACA AGC GCC G/3BHQ_1/-3′	100 nM
*RNASEP1* Forward	5′-CCA AGT GTG AGG GCT GAA AAG-3′	200 nM
*RNASEP1* Reverse	5′-TGT TGT GGC TGA ACT ATA AAA GG-3′	200 nM
*RNASEP1* Probe	5′-/Cy5/CCC CAG TCT CTG TCA GCA CTC CCT TC/3BHQ_2/-3′	100 nM

**Table 2 tropicalmed-08-00151-t002:** Criteria to analyze samples for *lip*L32 gene detection by qPCR.

Reaction	*lip*L32 Ct	*RNASEP1* Ct
*lip*L32 Control	<35	>40 or not detectable
*RNASEP1* Control	>35 or not detectable	<40
Positive Sample	<35	<40
Negative Sample	>40 or not detectable	<40
Inconclusive Sample		>40 or not detectable

**Table 3 tropicalmed-08-00151-t003:** Demographics of the population that the samples analyzed in this study originated from.

Sample Demographics	N (%)	Median (1st–3rd IQR)
**Sample size**	391 (100)	
**Sex ***		
Male	250 (63.94)	
Female	140 (35.81)	
Not declared	1 (0.26)	
**Age, Years**	385 (98.5)	36 (24–49)
**Race ***		
Black	33 (8.44)	
Brown	68 (17.39)	
White	128 (32.74)	
Yellow	40 (10.23)	
Not declared	122 (31.20)	

* Self-declared answers during admission.

**Table 4 tropicalmed-08-00151-t004:** Baseline characteristics of the sample-processing intervals up to qPCR reaction.

**Time Intervals**	**N (%)**	**Median, Days (1st–3rd IQR)**
From onset of symptoms to sample collection, days	356 (91.04)	4 (1–6)
From sample collection to DNA extraction	389 (99.49)	9 (7–14)
From DNA extraction to qPCR reaction	387 (98.98)	3 (1–5)

**Table 5 tropicalmed-08-00151-t005:** Sample collection time post-symptoms.

Result for *lip*L32 Detection	N (%)	Median (1st–3rd IQR)
Positive	189 (52.94)	3 (1–6)
Negative	157 (47.06)	4 (1–7)

**Table 6 tropicalmed-08-00151-t006:** Race demographics among tested samples.

Sample Demographics	N (%)	Positivity (%)
**Race ***		
Black	33 (8.44)	12 (36.36)
Brown	68 (17.39)	45 (66.18)
White	128 (32.74)	37 (28.91)
Yellow	40 (10.23)	8 (20.00)
Not declared	122 (31.20)	72 (59.01)

* Self-declared race at admission according to Brazilian Healthcare System (SUS) guidelines.

## Data Availability

The data set produced by the current study are available from the corresponding author upon request.

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
