# Peer review of "Effect of Demographics and Time to Sample Processing on the qPCR Detection of Pathogenic Leptospira spp. from Human Samples in the National Reference Laboratory for Leptospirosis, Brazil"

_tropicalmed, 2023, doi:10.3390/tropicalmed8030151_

Round 1

Reviewer 1 Report

Dear Sir 

Good Day

Page(s)                                     Comment                                         

69,72,73,74,75,84,85,90                   errors in formatting the pdf

126                                          re-write the highlighted

165                                          31.20%

214                                          write the missed reference

Follow the notes on the pdf.

References                    

Out of 27    14(2018-2022)

Write References in One style.

381    Abbreviated Journal; Int J Enteric Pathog

384    Full Journal;  Journal of infection and public health

386    Full Authors

395    Abbreviated Authors; Podgorek, Dasa, et al. 

Author Response

The point-by-point response to the reviewer is uploaded as a PDF file.

Reviewer 2 Report

Title:

Better to make it shorter

Introduction:

There are grammar issues and punctuation mistakes in the introduction and all around the article which have to solve.    

 Methods and Materials:

Please check out the whole article. There is misusing the term in the text. Some of them are exampled below, in brief:

 Line 94: “QIAGEN Quantinova Probe MasterMix “name the country and city for all similar materials and machines in the article

Line 96: Primer and probe sequences were prepared as described by [7] ??? please name the author then reference the article

Line 101: Ultrapure RNAse- and DNAse-free water do you mean UltraPure™ DNase/RNase-Free Distilled Water? use it correctly

 Line 103: “a rnaseP positive control” do you mean RNase Positive control?  if yes please correct it

 Line 105: “ CT was determined “ is not correct “ Ct was determined”

 Line 107 and 108:” In the analysis of sensitivity curves and determination of the detection threshold, samples were run as duplicates. For the routine diagnosis, each sample was analyzed without replicates.” How do you overcome the possible bias resulting from false positive and negative reactions in tested samples?

 Line 110-115: The whole paragraph is poorly organized and a bit confusing. I think better to put the threshold on a small table.

 Line 131: “p-value” should be the italic and upper case “P-value

 Results

Line 139-140: “reagent by MAT “I prefer “positive with MAT”

Line 213-214: “Diagnosis by MAT is usually recommended from the end of the first week throughout the second and third weeks (REF).” Reference????

 Line 242: “No significant differences were found among these groups (Figure 2b)” put the P values in the text (here and other parts), please

Line 266: “DNA processing procedure, with medians of 4 and 9, respectively “ Be precise about the numbers here and everywhere (in the Tables ). Do you mean 4 and 9 days?

Discussion:

It is good but needs a revision on writing as other parts of the article

Author Response

The point-by-point to the reviewer was submitted as a PDF file.

Reviewer 3 Report

the present study represents an interesting development in our understanding on the conditions for proper laboratory detection of leptospirosis.

The manuscript is written in OK English, with a few passages showing small slips of Romance-language phrasing, and a few passages (e.g. section 3.3, 3.4) appear to have been edited by someone other than the translator given some errors in syntax and grammar. A very light language revision should be considered to improve the manuscript, preferrably by someone besides the authors to facilitate picking up the issues in the text.

While I believe that the proposed structure of presenting the methods for analysis in each section of the results together with their results can be very effective on creating a meaning thread in the manuscript, I would personally advocate for organizing them in the methods to facilitate for readers to rethread the analytic logic of this study - which is rather sofisticated and complex - in special for the repeatibility and reproducibility.

However, I have to raise a few concerns on the study design:

1. There is an apparent lack of controls for confouders in the study. No controls for potential false-negatives on qPCR through the standard diagnostics method (MAT), no control for the effects of area of origin of the samples and/or socioeconomic status, which are known factors that affect risk of infection and overall positivity (See the works of Costa et al. 2014, 2015 for Salvador). This could be biasing the results of the tests, in special the ones involving socioeconomic information (such as racial profile and sex). I would like to know whether control measures were implemented to avoid these sources of bias, and if not, what bases the assumption that such aspects are not necessary to control.

2. Section 3.1 is not properly tied to the discussion, and appears to be "floating" with no proper discussion. In special, there is no explanation on why would MAT-negative samples be the only reactive ones for qPCR. There is no discussion on potential phenomena that could explain this particular issue in the detection. This compounds with the lack of described controls for the potential confounders.

Minor issues include:

On line 214, there is a reference cited as "(REF)", I believe that a reference is missing in this passage.

On section 3.5, I believe that a reference regarding the increased incidence during floods requires a reference. The sentence that requires the reference appears to be incomplete as well.

The P-values for the tests in sections 3.3 (first paragraph) and 3.5 are missing.

Author Response

(The authors gave the same response as above.)

Round 2

Reviewer 2 Report

Good job, please use the same format for the p-value in the text. It should be in italics and the numbers have to be in the same format (for example, P=0.004). Numbers are sometimes in another format in the text (P=.004).